# Mutual Effect of Gypsum and Potassium on Nutrient Productivity in the Alfalfa–Grass Sward—A Case Study

**DOI:** 10.3390/plants12122250

**Published:** 2023-06-08

**Authors:** Waldemar Zielewicz, Witold Grzebisz, Maria Biber

**Affiliations:** 1Department of Grassland and Natural Landscape Sciences, Poznan University of Life Sciences, Wojska Polskiego 28, 60-637 Poznan, Poland; waldemar.zielewicz@up.poznan.pl; 2Agricultural Chemistry and Environmental Biogeochemistry, Poznan University of Life Sciences, Wojska Polskiego 28, 60-637 Poznan, Poland; maria.biber@up.poznan.pl

**Keywords:** total yield, subsequent cuts, macronutrients—cations, micronutrient—cations, nutrient content, nutrient accumulation, total yield prediction

## Abstract

It was assumed that the production of alfalfa in soils naturally poor in available nutrients, such as potassium (K) and calcium (Ca), depends on the use of fertilizers. This hypothesis was validated in an experiment with an alfalfa–grass mixture carried out in 2012, 2013 and 2014 on soil formed from loamy sand that had a low content of available Ca and K. The two-factor experiment consisted of two levels of applied gypsum as a source of Ca (0, 500 kg ha^−1^) and five levels of PK fertilizers (absolute control, P60K0, P60K30, P60K60 and P60K120). The total yield of the sward was determined by the main seasons of alfalfa–grass sward use. Gypsum application increased the yield by 1.0 t ha^−1^. The highest yield of 14.9 t ha^−1^ was obtained on the plot fertilized with P60K120. Based on the nutrient content in the sward, it was shown that the main yield predictor was the content of K in the first cut of sward use. The reliable yield predictors, based on the total accumulation of nutrients in the sward, turned out to be K, Mg and Fe. The nutritional quality of the alfalfa–grass fodder, based on the K/Ca + Mg ratio, depended mainly on the season of the sward use, which was substantially deteriorated by the K fertilizer. Gypsum did not control this process. The productivity of the nutrients taken up by the sward depended on the accumulated K. Its yield-forming effect was significantly limited by manganese deficiency. The use of gypsum positively affected the uptake of micronutrients, consequently increasing their unit productivity, especially of manganese. Optimization of the production of alfalfa–grass mixtures in soils poor in basic nutrients requires micronutrients to be taken into account. Their uptake by plants can be limited by high doses of basic fertilizers.

## 1. Introduction

The growth of the human population is closely related to the growing demand for food [1]. According to Engel’s law, the structure of food expenditure also changes with the increase in household income. The forecast in this regard indicates a significant increase in demand for meat and dairy products [2]. The production of milk and especially meat is carried out extensively, as exemplified in Argentina, Australia, Brazil and the U.S.A. Any increase in this production system depends on the area of pastures [3,4]. Alternative intensive production systems require two basic inputs, water and nitrogen. A side effect of this system is the increased threat to the environment, which results both from the consumption of non-renewable resources (natural gas) for the production of nitrogen fertilizers and the pollution of water and air [5].

An alternative to intensive milk and meat production is the cultivation of legumes in their pure state or in mixtures with grasses. This production system fully meets the requirements of sustainable agriculture. The basic assumption of this food production strategy is to optimize the use of production resources and, at the same time, to minimize the negative impact on the environment [3,6]. In a ruminant feed production system where alfalfa predominates, the source of nitrogen is the atmosphere. Therefore, this legume crop should be considered as the primary source of N in the farm soil/plant system [7]. In the mixture of alfalfa with grasses, the source of N for the companion species, i.e., grasses, is both soil and its transfer from alfalfa [8].

Alfalfa is grown for ruminant feed in all regions of the world excluding the tropics. Optimum temperatures fall within a wide range, from −5 °C (night frost) to +20 °C. In rainfed agriculture, optimal rainfall is in the range of 300 to 600 mm per year [9]. For this reason, production of this crop and its mixtures with grass species is widespread in temperate regions of the world [10,11]. Moreover, alfalfa has a naturally large root system and grows deeply into the soil to a depth of several meters. For this reason, this crop can be grown in regions with frequent droughts or low rainfall during the growing season, down to 200 mm [9,12]. A deep and extensive root system allows a plant not only to take up water from the deeper soil layers, but also to absorb nutrients efficiently. Alfalfa persistence and production in unfavorable environments can be also explained by the high activity of microorganisms in the rhizosphere [13,14].

One of the most important characteristics of alfalfa is its natural ability to grow and be produced in a wide range of environmental ecosystems, as related to soil fertility and water availability. The basis of alfalfa adaptability to different environments is naturally high soil reaction, and good resources of potassium (K) and calcium (Ca) [15,16,17]. K is an essential nutrient for the transport of N compounds formed in the root as a result of N_2_ fixation in the root nodule. For this reason, K can be regarded as an important nutritional factor determining the agricultural use of alfalfa’s N_2_ fixing potential [18,19]. On this basis, it can be concluded that alfalfa is sensitive to a good K supply, regardless of the source, as the nutritional factor affecting the yield of the sward. Plants well supplied with K show a higher resistance to abiotic stresses, such as shortage of water [20]. The second crucial nutrient for alfalfa is Ca, which is required for both the formation of plant tissue, including roots, as well as N_2_ fixation [21]. The high tolerance of this plant to relatively unfavorable conditions is probably due to its high requirements for Ca. This nutrient plays a crucial role in plant defense-signaling routes, which is caused by water stress [22]. A deficiency of available Ca in soil cropped with alfalfa not only reduces the number of root nodules, but also reduces their size, resulting in a lower amount of fixed N_2_ [23]. Therefore, the largest and most stable yields of alfalfa are obtained when it is grown in fertile soils with an optimal supply of water. Under these conditions, during the growing season, this crop fixes a large amount of N, which ranges from 300 to 600 kg N ha^−1^ [17,24].

Another important aspect of alfalfa nutrition with K, apart from the yield, is feed quality for ruminants. According to the nutritional standards for this group of animals, the content of K, Ca and magnesium (Mg) is around 1.0%, 0.62% and 0.19% DW, respectively [25]. The optimal ranges of the content of these minerals in fodder from grasslands in Poland is 1.7–2%, 0.7% and 0.2% DW, respectively [26]. The content of these three minerals in alfalfa plants at the optimal harvest time (beginning of flowering) is assuredly higher. According to Marković [27], the content of K frequently exceeds 2% DW, calcium reaches 1% DW, and Mg can reach 0.4% DW. The classic indicator for assessing the content of minerals in feed for animals is the K(Ca + Mg) ratio; 2–2.2 is considered optimal [25,28]. Based on the this set of data, it can be stated that these conditions are met for grassland fodder in Poland, as presented above, but are too narrow for alfalfa [26,27].

Soils formed from post-glacial mineral materials are naturally poor in alkaline cations and microelements. Such soils dominate in the northern part of Europe, including Poland [29]. In such soil conditions, it could be expected that alfalfa would respond positively to the application of K and/or Ca fertilizers. The results of research on K fertilization are not unambiguous. Some studies have shown alfalfa to respond significantly to the application of nutrients, including K [15,24]. The objective of the study was to evaluate the effect of the applied Ca in the form of gypsum and increased doses of K fertilizer in soil poor in the available forms of both nutrients on the productivity of nutrients accumulated in the alfalfa–grass sward.

## 2. Results

### 2.1. Content of Main Cations and the Yield of Alfalfa–Grass Sward

The total yield (TY) of the alfalfa–grass sward significantly depends on the years and experimental factors, but no interaction between them was found (Table 1). The highest yield was recorded in the second main year of the sward utilization (2013) (Figure 1). The average sward yield this season was higher by 3.6 t ha^−1^ (31%) compared to the first main year of use (2012)*,* and 3.0 t ha^−1^ (23.8%) higher compared to the third year. The application of gypsum and PK fertilization averaged over the years resulted in a TY increase of 1.0 t ha^−1^ (7.8%). However, the greatest yield increase resulted from PK fertilization. The TY of the alfalfa–grass sward in response to the applied phosphorus increased by 1.3 t ha^−1^ compared to the absolute control (AC, plot without P and K). In response to increasing doses of fertilizer K, TY was consistent with the quadratic regression model:(1)$$\mathrm{TY}=-0.00017K^{2}+0.039K+12.6\;\mathrm{for}\;n=30,{\;R}^{2}=99,\;p\;\leq 0.001\;$$

This model clearly indicates that the maximum TY of 14.9 t DW ha^−1^ was obtained for the optimal K dose of 114.7 kg K ha^−1^.

The TY of the alfalfa–grass sward, as results from the conducted stepwise regression, was determined by the content of two nutrients, i.e., K in the first cut of the sward (K1) and negatively with the content of magnesium in the third cut (Mg3):(2)$$\mathrm{TY}=12.1+0.32K1-\mathrm{Mg}3,\;\mathrm{for}\;n=30,{\;R}^{2}=0.81,\;p\;\leq 0.01$$

The TY, as seen in the results from this equation, was significantly affected by the increasing content of K in the first cut and decreased with an increase in the Mg content in the third cut of the alfalfa–grass sward. The content of both nutrients was significantly dependent on the GYP × POT interaction (Table 1). The K content was within a narrow range of 21–25 mg kg^−1^ DW. The content of K in the sward in GYP–0 (gypsum control) showed a stable level on the plots with no or a low dose of K, while its significant increase was recorded for the first time in the plot with 60 kg K ha^−1^ (Table 1). Nevertheless, the trend of K content on plots fertilized with K was consistent with the liner regression model (Figure 2, R^2^ = 0.88). A slightly different tendency was observed in the GYP–500 plot (gypsum fertilized plot). In plots with no or a low dose of K, a progressive increase in K content was observed, reaching stabilization in the plot with 30 kg K ha^−1^. The trend of K content increase on this object is best reflected by the quadratic regression model (Figure 2, R^2^ = 0.93).

A completely opposite tendency was observed for the content of Mg in the alfalfa–grass sward (Figure 3). In general, the Mg content ranged from 2462 mg kg^−1^ DW in plots fertilized with the highest doses of K, up to 3050 mg kg^−1^ DW in the control K. The content of Mg in the sward on GYP–0, despite a small variability, was highly stable in plots without or with a low K dose. It decreased sharply in plots fertilized with the K dose of 60 and 120 kg K ha^−1^. An almost identical trend was obtained in GYP–500. However, a significant decrease in Mg content was observed in all the plots fertilized with K compared to the K control (K0). The trend of Mg content in both objects can be best described by the linear regression model.

The TY of the alfalfa–grass sward also showed a significant negative relationship with the Ca content in the first cut (Ca1). Its content was negatively correlated with the K content in the sward of the first cut (K1), but at the same time, it was positively correlated with Mg3 (Table A1). The content of Ca displayed a small variability of about 700 ± 10 mg Ca kg^−1^ DM in the first main year of the sward use (Appendix A). In the second year, a significant decrease in Ca content, from 7683 Ca kg^−1^ DM in the AC to 4137 Ca kg^−1^ DM in the plot with highest dose of K, was observed. In the third year, its content in the AC plot exceeded 1% of DM (12,226 Ca kg^−1^ DM). Nevertheless, a significant decrease was observed in accordance with the increased dose of applied K (6785 Ca kg^−1^ DM in the plot with 120 kg K ha^−1^).

The K/(Ca + Mg) index was significantly correlated with the TY (Table A1). The level of TY prediction, based on CR1, was the same as that obtained for K1:1.The content of K in the sward of the first cut (K1):
(3)$$\;\mathrm{TY}=0.45K1+3.19,\;\mathrm{for}\;n=30,{\;R}^{2}=0.73,\;p\;\leq 0.001\;$$

2.The K/(Ca + Mg) ratio:


(4)
$$\;\mathrm{TY}=2.33\frac{K}{\mathrm{Ca}+\mathrm{Mg}}+7.8\;\mathrm{for}\;n=30,{\;R}^{2}=0.71,\;p\;\leq 0.001\;$$


Taking into account that the Ca1 was significantly and negatively correlated with K1, the course of both regression models indicates the dominant impact of the numerator on the obtained indices. The effect of the K fertilizer dose on the index was significantly variable in the subsequent years of the study (Figure A1). In the first main year of using the sward, the standard value of this index in the range of 2.0–2.2 [25,28], averaged over fertilization treatments, reached 1.8 and 2.2 on the gypsum control and GYP–500, respectively. The most stable ratio, regardless of the gypsum use, was obtained in the P60K60 plot. In the second main year, the index values significantly exceeded the standard ratio. Moreover, they showed a progressive increase in accordance with the increased doses of K fertilizer. The only exception was the AC in the GYP–0 plot. The respective averages for the gypsum main plots were 3.2 and 3.0, respectively. In the third main year, the respective averages were much lower, and reached 2.1 and 2.2. The standard nutrient ratio was exceeded mainly in the GYP–500 plot fertilized with K.

### 2.2. Accumulation of Main Cations in Alfalfa–Grass Sward

The TY of the alfalfa–grass sward was significantly correlated with most of the characteristics related to the total amount of accumulated nutrients (Table A2). For their total accumulation (T), the highest value of the correlation coefficient was found for K, followed by Mg:(5)$$\mathrm{TY}=0.023\mathrm{TK}+6.62\;\mathrm{for}\;n=30,{\;R}^{2}=0.84,\;p\;\leq 0.001$$
(6)$$\mathrm{TY}=0.29\mathrm{TMg}+2.72\;\mathrm{for}\;n=30,{\;R}^{2}=0.80,\;p\;\leq 0.001$$

The obtained linear models clearly indicate that each increase in the amount of both nutrients in the sward resulted in a progressive increase in the TY. Moreover, both traits were significantly correlated with each other (r = 0.78). The amount of Ca also had a positive impact on the TY, but this relationship was poor (r = 0.42) (Table A2).

The analysis of the total K accumulation (TK) by the alfalfa–grass sward clearly showed its dependence on the Y × POT interaction (Table 2). In the first main year of the sward use, the TK increased progressively with the doses of applied P and K fertilizers (Figure 4). It should be emphasized that TK was significantly dependent on K accumulation in the first cut of the sward (r = 0.88). The amount of TK in the plot with 120 kg K ha^−1^ was ^2^/_3_ higher than in the AC plot. This value was significantly lower compared to the second main year in the AC plot. In this particular year, the increase in the TK was progressive with the applied doses of fertilizer K. The TK obtained in plots fertilized with the highest doses of K fertilizer was almost three times greater (2.77-fold increase) compared to the baseline value in the AC plot in the first main year of the alfalfa–grass sward use. In the third year, the trend of TK was the same as in the previous year, but the values were about 100 kg K ha^−1^ lower.

The general pattern of total Mg accumulation (TMg) by the alfalfa–grass sward also depended on the Y × POT interaction (Table 2). However, the TMg variability in response to PK doses in subsequent seasons was much lower (Figure A2). In the first season, a net increase in TMg to the K dose of 30 kg ha^−1^ was found, which then underwent stabilization. The increase in TMg due to PK application compared to AC was almost 28% (+7 kg Mg ha^−1^). In the second main season, TMg, as in the case of TK, exceeded the highest values recorded in the first year, even in the AC plot. However, the recorded increase resulted only from the application of P. In the third main season, all the TMg values were lower by about 10–12 kg Mg ha^−1^, as compared to the previous season.

The highest value of the correlation coefficient between the amount of an examined nutrient and the TY was recorded for Mg in the first cut (Mg1a, r = 0.91) (Table A2). A slightly lower strength of the analyzed relationships was obtained for K accumulated in the third cut (K3a, r = 0.88). The stepwise analysis allowed us to distinguish the following set of TY indicators:(7)$$\mathrm{TY}=1.86+0.02K1a+0.026K2a+0.39\mathrm{Mg}1a+0.21\mathrm{Mg}3a\;\mathrm{for}\;n=30,{\;R}^{2}=0.96\;\mathrm{and}\;p\;\leq 0.01$$

The first dominant component of Equation (7) is K1a. Its variability resulted from the Y × GYP *×* POT interaction (Table 2). The effect of the PK treatments was slightly different in the first season compared to the other two (Figure A3). In the first season, K1a was much lower (about 40%) compared to the values in the other two seasons. However, the effect of gypsum, averaged over PK treatments, was much higher. The plants treated with gypsum accumulated 27.5% more K (+15.4 kg K ha^−1^). In the second season, despite the higher accumulation of K, the difference between the gypsum plots was smaller and amounted to only 5.9% (+6 kg K ha^−1^). In the third season, it was 13.2% (+12.5 kg K ha^−1^). In the second season, gypsum had a negative impact on the amount of K in the AC plot, whereas it positively affected the fertilized ones. In the third season, this negative tendency was observed in the AC and K0 plots. However, a strong increase was noted in the plots fertilized with K.

The second component of Equation (7), i.e., K2a, showed a significant dependence on the Y × POT interaction (Table 2). In the first and the third seasons, a progressive impact of POT treatments was observed on the accumulation of K in the sward (Appendix A). The amount of K was significantly higher in the first season and was almost at the same level in the remaining seasons. The effect of gypsum was negligible. The highest K2a values were associated with the highest doses of K fertilizer. In these plots, a slight decrease in K2a was noted in the third season.

The third and fourth components of Equation (7), related to the accumulation of Mg in the first and third cuts of the alfalfa–grass sward, showed a response to all the studied factors, but not to the interaction between them (Table 2). The amount of Mg in the first cut of the sward (Mg1a) was slightly higher in the second season compared to the other two seasons. The impact of gypsum was significant and resulted in a net accumulation of Mg of 10.2% (+1.1 kg ha^−1^). The effect of the PK treatments was progressive up to a K dose of 60 kg ha^−1^. Compared to the AC, Mg1a increased by 22.5% (+2.3 kg ha^−1^). Mg accumulation in the third cut of the sward (Mg3a) was much more diverse in the subsequent seasons. In the second season, it was 71.6% (+7.6 3 kg ha^−1^) and 95.6% (+8.9 3 kg ha^−1^) higher than in the first and third seasons, respectively. The effect of gypsum was positive, but slightly less compared to the first cut. The effect of the PK treatments was significant between treatments without and with potassium. The application of K significantly increased the amount of Mg in the sward.

The amount of Ca in the alfalfa–grass sward, despite its high variability, especially in the second and third cut, did not significantly affect the TY. In the first two cuts, the highest amount of Ca was recorded in the third season. The most pronounced effect of gypsum was observed in the second cut. The amount of Ca in the sward increased by 19.3% (+7.3 kg ha^−1^). The effect of PK was different in subsequent cuts. In the first and third seasons, the increased doses of PK caused a decrease in the amount of Ca in the sward. In the second cut, it was the highest, but the effect of PK was positive only for the highest doses of K.

### 2.3. The Content of Micronutrients

The content of micronutrients in the sward of the alfalfa–grass mixture depended, to the greatest extent, on years, and to a lesser extent on experimental factors (Table 3). The average Fe content in the sward was 42.4 ± 2.3 mg kg^−1^ DM and it showed a downward trend in successive cuts. The content of Fe in each cut varied greatly between seasons: while there was no significant change between cuts during 2012, there was a decrease during 2013 and an increase during 2014.

The average content of Mn was 24.1 ± 2.5 mg kg^−1^ DM, and during both 2012 and 2013, Mn was highest in the second cut; however, in 2013, as with Fe, it was highest in the third cut (Table 3). The observed trends were fully confirmed by the relationships between Fe and Mn (Table A3). These associations were significant and positive for the first and second cuts, and negative for the third cut. The most variable trend in the successive cuts of the sward was observed for zinc. Its content was the highest in the first cut, then significantly decreased (21.3 ± 4.0 mg kg^−1^ DM). Moreover, it was the only micronutrient whose content did not show a significant relationship with the TY (Table A3).

The average content of Cu (4.4 ± 0.3 mg kg^−1^ DM) was also assessed. The Cu content was highly variable among the cuts. It was lowest in the second cut in 2012 and the third cut in 2013, while in 2014, it was higher. The use of gypsum resulted in a significantly lower Cu content in the sward of the first cut.

The TY of the alfalfa–grass showed a weak relationship with the content of the studied micronutrients in the sward (Table A3). Across 12 traits, only five were significantly correlated with the TY. A positive relationship was noted for the content of Fe in the first and second cuts (Fe1, Fe2). Negative relationships were obtained for the content of Mn in the third cut (Mn3), and Cu in the first and third cuts (Cu1, Cu3). A stepwise regression analysis indicated Cu3 as the key predictor of the TY:(8)$$\mathrm{TY}=19.7-1.52\mathrm{Cu}3\;\mathrm{for}\;n=30,{\;R}^{2}=0.33\;\mathrm{and}\;p\;\leq 0.001$$

Equation (8) indicates that each increase in the Cu content in the third cut of the alfalfa–grass sward resulted in a significant decrease in the TY. Cu3 was negatively correlated with Fe1 and Fe3, which had a positive impact on the TY. The same relationships were observed for Mn1 and Mn2 (Table A3). An inverse and strong relationship was noted between Cu3 and Mn3 (r = 0.75). Cu3 was significantly dependent on the Y × GYP × POT interaction (Appendix A). The greatest variability was noted in the first season in the GYP–0 plot. The highest Cu content of 6 mg kg^−1^ DW was recorded in the P60K30 plot. The plants fertilized with gypsum showed a progressive decrease in Cu content with increasing PK doses. In the second season, a slightly lower content of Cu was generally recorded in the plants grown in the GYP–500 plot. In the third season, this general trend was also evident, but the differences between the PK treatments were insignificant.

### 2.4. Accumulation of Main Micronutrients-Cations in Alfalfa–Grass Sward

The total accumulation of three of the four analyzed micronutrients was significantly correlated with the TY (Table A4). The only exception was Mn. The total amount of Fe (TFe) in the sward of the alfalfa–grass mixture had the greatest impact on TY:(9)$$\mathrm{TY}=0.016\mathrm{TFe}+4.77\;\mathrm{for}\;n=30,{\;R}^{2}=0.82\;\mathrm{and}\;p\;\leq 0.001$$

This equation clearly shows that 82% of the TY variability was due to variability in the TFe, which was weakly, but positively correlated with TZn and Tcu, but not with TMn (Table A4). The TFe was driven by the GYP × POT interaction (Table 4, Figure 5). The positive impact of gypsum was significant in plots without K fertilizer. In the plots with intermediate K doses, a slightly higher TFe was recorded in the GYP–0 plot. A stepwise regression analysis of all the cuts of the alfalfa–grass sward revealed a broader set of traits responsible for TY prediction:(10)$$\mathrm{TY}=-0.02+0.02\mathrm{Fe}2a+0.07\mathrm{Mn}1a+0.06\mathrm{Zn}3a+0.16\mathrm{Cu}1a\;\mathrm{for}\;n=30,{\;R}^{2}=0.90\;\mathrm{and}\;p\;\leq 0.001$$

Zinc accumulated in the third cut of the alfalfa–grass sward had the greatest single impact on the TY (Table A4). Moreover, it was significantly correlated with the amount of remaining micronutrients, showing the highest correlation coefficient with Cu in the third cut (r = 0.87). The general trend of Zn3a was significantly dependent on the season (Appendix A). In the first season, Z3a was the lowest, but the response to gypsum was the highest, reaching 18.2% (+12 g ha^−1^). In the GYP–500, a progressive increase in Zn3a was observed in line with the increased PK doses. In the second season, Z3a reached at least twice higher values, but no effect of gypsum was noted. The high, almost stable Zn accumulation in the GYP–0 plot, regardless of PK treatments, should be emphasized. On the other hand, its high variability was observed in GYP–500, mainly due to the exceptionally high amount of Zn in the sward in the P60K60 plot. In the third season, Zn3a was even lower than in the first season. A positive impact of PK fertilizers was observed and was more pronounced in the GYP–0 plot.

The Mn content in the first cut of the alfalfa–grass sward responded to all the tested treatments, but no interaction between them was found (Table 4). The highest value of Mn1a was recorded in the second season. The effect of gypsum was significant, reaching 13.2% (+10 g ha^−1^). The application of P fertilizer was sufficient to reach a significant increase in Mn1a compared to the AC plot.

The first component of Equation (10), i.e., the amount of Fe in the second cut (Fe2a) of the alfalfa–grass sward, showed high variability, determined by all the factors tested (Appendix A). The effect of gypsum was most evident in the first and third seasons. In the first season, gypsum increased the Fe accumulation by 9.2% (+19 g ha^−1^). Moreover, it responded progressively to PK fertilization. The most spectacular increase in Fe was found in the AC plot and the plots fertilized with the highest doses of K. In the second season, the average Fe2a value was at the level recorded in the previous season. The effect of PK depended on the use of gypsum. In the GYP–0 plot, a significant increase was found in response to PK, whereas in the GYP–500 plot, it was significantly variable. In the plots without or with low doses of K fertilizer, a decrease below AC was noted. In the third season, the amount of Fe in the alfalfa–grass sward was the lowest, but it responded significantly to gypsum use (17%, +23 g ha^−1^). The effect of PK was more pronounced in the GYP–500 plot.

The last component of Equation (10), i.e., the amount of Cu in the first cut of the sward (Cu1a), responded to the Y × POT interaction. In the first two seasons, a slight positive trend towards increased application of PK fertilizers was observed (Appendix A). In the third season, the amount of Cu in the sward increased up to a K dose of 30 kg K ha^−1^, and then stabilized.

### 2.5. Nutrient Unit Productivity—NUP

Only four of seven NUP indices showed a significant association with the TY (Table A5). The strongest values of the correlation coefficients were noted for Mn and K and were significantly lower for Fe and Cu. However, the stepwise regression analysis performed clearly pointed to FeUP and MnUP as the main predictors of the TY:(11)$$\mathrm{TY}=11-0.24\mathrm{FeUP}+0.044\mathrm{MnUP}\;\mathrm{for}\;n=30,{\;R}^{2}=0.87\;\mathrm{and}\;p\;\leq 0.001$$

The obtained regression model shows that the unit productivity of Mn accumulated (MnUP) in the alfalfa–grass mixture was too low and limited the TY. This index was driven by the GYP × POT interaction (Table 5, Figure 6). Its values, regardless of the use of gypsum, were low in the plots without K fertilizer. Its significant increase was first noted in plots fertilized with K fertilizer. The use of gypsum resulted in a much higher productivity of Mn taken up by the alfalfa–grass sward. The FeUP indices also depended on the GYP × POT interaction, but the resulting trends were less clear (Appendix A). In the GYP–0, the application of K fertilizer resulted in a significant decrease in FeUP. In the GYP–500, the indices were much higher as compared to GYP–0, and the negative impact of K fertilizer was less pronounced.

The unit productivity index of K (KUP) showed a significant and positive relationship with the TY (r = 0.73 ***) (Table A5). Moreover, it was strongly associated with MnUP (r = 0.75 ***) and was significantly weaker with CaUP (r = 0.24 *). This index was driven by the Y × POT interaction (Table 5, Appendix A). The highest KUP was found in the first season and was significantly lower in the other two seasons. The most pronounced differences between the production seasons were observed in the AC plot. In the 2012 season, the value of this index was 37.8%, which was 19.2% higher compared to the 2013 and 2014 seasons. As a rule, KUP decreased with the increase in the doses of K fertilizer. For the highest K dose, the differences were almost the same, reaching −28 and −25% in 2013 and 2014, respectively. CaUP followed the same trend as noted for KUP, but a progressive decrease in its value was observed in subsequent seasons. ZnUP showed a reverse seasonal trend as compared to CaUP (Table 5). Despite this fact, both indices were significantly correlated with each other (Table A5). MgUP, which reached the highest values among the examined indices, was not significantly related to either the TY or other UP indices. The CuUP indices were also high but showed poor correlation with the TY, CaUP and MnUP.

## 3. Discussion

### 3.1. Yield of Alfalfa–Grass Sward

The total yield of the grass–legume mixture was high, reflecting the optimal precipitation during the study period. The amount of precipitation during the growing season was in the optimal range, ranging from 373 mm in the second main season to 561 mm in the first season. The total yield harvested in the second main season was in line with a typical trend in alfalfa productivity in the successive season of sward use [30,31,32]. The content of available of K and Ca in the soil before the experiment was insufficient for a crop such as alfalfa (Table 6). The maximum K dose of 120 kg K_2_O ha^−1^ in relation to the initial level of soil fertility was slightly below the fertilization recommendations in the U.S.A. for alfalfa (150 kg K_2_O ha^−1^) [33]. Its total amount applied during these four seasons for the plot with maximum use was 480 kg K_2_O ha^−1^. The total amount of Ca applied as FGD–gypsum was 426 Ca ha^−1^. The efficiency of the fertilizers was 60% for K and 26% for Ca.

The effect of FGD–gypsum on the sward yield was significant, indirectly highlighting the effect of the supplied nutrients, i.e., Ca and S, to soil poor in their content [34,35]. However, its indirect effect through enhanced microbial activity cannot be ignored [14]. The maximum yield of the sward resulted from the interaction of the P and K fertilizers. The application of P fertilizer caused an increase in the alfalfa–grass mixture yield compared to the absolute control by 11% (+1.2 ha^−1^). The importance of P in the nutrition of legumes is well documented [36]. A positive impact of applied P fertilizers on the number and size of nodules and the increased content of N in alfalfa has been clearly demonstrated [37]. A further increase in the yield of the alfalfa–grass mixture was driven by progressive doses of applied K. The increase in the total yield (TY) by 17.3% (+2.3 t ha^−1^) in the plot fertilized with 120 kg K_2_O ha^−1^ as compared to the K control indicates that K was the key nutrient limiting the growth and productivity of the grass–legume mixture in the test site. The obtained TY of 14.9 t DW ha^−1^ was harvested after applying only of 115 kg of K_2_O ha^−1^. The level of the TY was high, indirectly highlighting the optimal environmental and soil conditions for the growth and productivity of the alfalfa–grass sward. This K rate is very often considered too small to obtain a sufficiently high yield and K content [38]. The same level of yield was presented in Macolino et al. [39]; however, 300 kg K_2_O ha^−1^ was used in their research. The authors of the current study found no response of alfalfa to P. The effect of the interaction of both nutrients applied in quantities of 50 kg P and 300 kg K ha^−1^ on yields of alfalfa was noted in the studies carried out in India [40].

The dominant role of K on the productivity of the alfalfa–grass sward was only partly confirmed by its utilization efficiency. The values of this index, presented as the potassium unit productivity, decreased in both the subsequent years of the study and even more strongly in response to the increased doses of applied K fertilizer. This coincidence indicates a decrease in the yield-forming role of K resulting from its systematic application. The limiting effect of Ca content on the productivity of K was small, with manganese being the most important.

### 3.2. The Content of Macronutrients vs. Total Yield

The production of fodder for ruminants causes a serious conflict of interest for the farmer. This is due to the fact that farmers mainly focus on maximizing the yield, which in turn often leads to a deterioration in the nutritional quality of the feed [15,41]. In the presented study, the first cut of the sward was of key importance for a reliable assessment of the impact of the K fertilizer dose on the yield of the alfalfa–grass mixture, but not on its nutritional quality. The total yield of the alfalfa–grass sward, as discussed above, depended to a large extent on the applied dose of K fertilizer (Equation (1)). The negative effect of the Mg content, as well as the positive effect of its amount in the sward on the total sward yield, proves that Mg resources in the soil were not used efficiently by the alfalfa–grass mixture (Equation (7)). The high content of Mg in the sward acted as a nutritional buffer. This conclusion results from the fact that the decrease in the content of Mg in the sward as a result of increasing K doses did not result in a decrease in Mg productivity and yield.

The importance of K fertilizer on the yield was enhanced by the fact that the TY was reasonably predicted by the following:(1)K content in the first cut of the sward;(2)Total amount of K accumulated in the sward.

The content of K in the first cut of the alfalfa–grass sward was at the level of 22.8 ± 4.1 g kg^−1^ DM. This value does not exceed the standard range of K content in the alfalfa sward harvested before flowering (20.1–35 g kg^−1^ DM) [42]. This standard range was also not exceeded in the other two cuts. A progressive increase in the content of K in the sward was observed in line with the increase in the TY, regardless of the consecutive cuts. This type of the relationship between the yield of a crop and the content of a given nutrient clearly indicates that the rate of K uptake by plants in the studied sward preceded the rate of the alfalfa–grass biomass increase. At the same time, the dilution effect for the Ca content in the sward of the first cut was noted. This was in line with the negative relationship between the K and Ca contents. The opposite response of both nutrients to the increasing biomass of the alfalfa–grass mixture suggests the presence of antagonism exerted by K. In the remaining cuts of the sward, especially in the third one, the content of K was ahead of the biomass growth rate, similar to the first cut. At the same time, the content of Ca showed a low or no response to the increase in the yield.

An important relationship is the tetany ratio between K on the one hand and Ca and Mg on the other, and this is optimal in the range of 2.0–2.2 [25,28]. This nutritional index was slightly above 2.2 for the first two cuts of the alfalfa–grass mixture. In the third cut, it reached 3.3 ± 1.8. The trend of the K/Ca + Mg ratio, regardless the subsequent cuts of the sward, followed the power regression model, taking into account the content of Ca as the independent value. The calculated ranges for the amount of Ca required in the sward of the alfalfa–grass mixture for the first to the third cuts were 0.72–0.80, 0.68–0.78 and 0.60–0.66 kg^−1^ DM, respectively, indicating the highest requirements for Ca during the spring growth period.

The most important goal of gypsum application is to reduce the dilution effect of Ca. However, the main reason for the tetany ratio increase in the third cut was not the increase in the content of K, but the sharp decrease in the content of Ca and the slight decrease of Mg. In this discussion, changes in the tetany ratio in subsequent years of the alfalfa–grass sward cultivation cannot be overlooked. This index reached the highest values in the second main year of the sward use, exceeding 3.0. It should be highlighted that, in general, the increase in the index was not dependent on the experimental factors, but the overall rhythm of alfalfa growth. The increase in the tetany index in the second main season was most likely the result of the ingrowth of alfalfa roots in the deeper soil layers, expanding the root surface area on one side, and acidifying the rhizosphere on the other. These processes most likely led to an increased uptake of K [43]. Moreover, the increase in the index was stimulated by the application of K fertilizer and was progressive to its dose. This phenomenon was observed in the second and third main seasons. Ca taken up by plants in the sward showed a control effect, reducing the value of the tetany index, but its effect was weaker compared to that of K. This was probably due to the very low resources of available Ca in the deeper soil layers.

### 3.3. The Content of Micronutrients vs. Total Yield

The content of micronutrients in the sward of the alfalfa–grass mixture did not allow their use as a strong yield predictor. The variability of the characteristics of this sward resulted mainly from the seasonal factor, i.e., years and cut. The content of Fe and Mn revealed similar trends, showing a decrease in the first two cuts in subsequent years of use. A completely different relationship was noted in the third cut, where the highest values were recorded in the third main year. The trends obtained are consistent with the observations by Wang et al. [31]. The specific yield-forming functions of both nutrients has been confirmed by two facts:(1)Total accumulation of Fe among the tested micronutrients determined the yield variability to 88%;(2)The Mn unit productivity was the main nutritional factor among the seven tested nutrients, affecting the total yield of the sward of the alfalfa–grass mixture.

The predictive worth of the total amount of Fe in the sward was not lower than that of K and Mg (Equations (5), (6) and (9)). It should be highlighted that the best positive impact of the applied gypsum was revealed in the plots not fertilized with K fertilizer. The reduction of Fe accumulation occurred in plots with an intermediate level of applied K (30 and 60 kg K_2_O ha^−1^), where a significant increase in yield was observed. However, Fe did not limit the unit productivity of K. This nutritional factor was determined to be Mn. The limiting effect of Mn accumulated in the sward on the TY was revealed mainly in the first and the third years of its use. In addition, in each cut of the sward, Mn and Fe showed a positive relationship, both in terms of content and accumulation. This type of interdependence between both nutrients and the yield indicates their significant impact on the harvested yield of the alfalfa sward. At this point, the specific role of gypsum should be underlined. In regard to gypsum, an increase in the amount of micronutrients accumulated in the sward was observed. More important, however, is the fact that gypsum significantly increased the unit productivity of the micronutrients. This applies in particular to Mn, which significantly limited the unit productivity of K.

## 4. Materials and Methods

### 4.1. Research Site

A field experiment with a mixture of alfalfa and grass was conducted from 2011 to 2014 at the Brody Experimental Station, Poznan University of Life Sciences, Poland (52°44′ N, 16°28′ E) in soil formed from loamy sand, classified as Albic Luvisols (Neocambic) [44]. The presented study does not include the season in which the experimental field was set up, i.e., 2011. The content of organic carbon (C_org_) in the 0.0–0.3 m layer fluctuated during the study at the level of 12.8 ± 0.25 g kg^−1^ soil (losses on ignition). The soil pH was in the neutral range (1 M KCl), but increased gradually with the soil depth. The content of the available macronutrients, measured before the experiment started in 2011, was very high for magnesium (Mg) and high for phosphorus (P), but low for K and very low for Ca. The content of the available micronutrients, which were measured at the same time, was, in general, sufficient for pasture plants (Table 6).

**Table 6 plants-12-02250-t006:** The content of available nutrients in the soil profile in the experiment setup ^1^, mg kg^−1^ soil.

Nutrient		Mean	SD	CV.%	Nutrient	Mean	SD	CV. %
K ^2^	0–30	88.2 L	11.0	12.5	Fe ^4^	241.3 M	2.1	0.9
	30–60	84.0 L	12.9	15.4		220.1 M	22.9	10.4
	60–90	63.5 L	6.4	10.1		183.7 M	35.9	19.6
P ^2^	0–30	158.1 H	5.4	3.4	Mn ^4^	68.8 M	1.3	2.0
	30–60	159.9 H	6.6	4.1		65.5 M	6.9	10.6
	60–90	90.8 L	15.6	17.1		46.5 M	11.9	25.5
	0–30	164.0 VH	11.2	6.8	Zn ^4^	4.6 M	0.1	3.1
Mg ^2^	30–60	191.5 VH	61.1	31.9		4.1 M	1.1	26.2
	60–90	140.5 VH	39.1	27.8		2.3 M	0.6	24.6
	0–30	690.4 VL	121.1	17.5	Cu ^4^	3.3 M	0.2	6.5
	30–60	674.9 VL	17.5	2.6		2.9 M	0.2	7.4
Ca ^3^	60–90	450.7 VL	38.5	8.5		2.2 M	0.6	29.6

^1^ Mehlich 3 [45]; ^2,3,4^ availability classes: VL—very low; L—low; M—medium; H—high; VH—very high [46,47,48].

### 4.2. Weather Conditions

The basic data on the course of weather in the 2012, 2013 and 2014 growing seasons are presented in Table 7. The optimum environmental conditions for the growth of alfalfa, as the dominant plant in the alfalfa–grass mixture, are 10–20 °C and 300–600 mm of precipitation in the growing season [9]. In the first main growing season (2012), the weather was optimal for plant growth and the production of the alfalfa sward. The air temperature, and in-season distribution of precipitation were within the optimal range for alfalfa, as the dominant species in the sward. In the second main season (2013), the temperatures were slightly higher than in the first, but still in the optimal range. The total precipitation was just under 400 mm, but within the optimal range. The third main season (2014) was even warmer than the previous years, with just under 500 mm of precipitation. Attention should paid to the phenomena of excessive precipitation, i.e., rainfall, which was twice the long-term average. These events were recorded in June and July 2012, June 2013 and July 2014.

### 4.3. Experimental Design

The field experiment, arranged in a two-factor split-plot design and replicated four times, was begun in the summer 2011. The fore crop of a mixture of alfalfa (*Medicago sativa* L.) with two grass species was winter oilseed rape. The sowing mixture, composed of alfalfa—80%, meadow fescue (*Festuca pratensis* Huds.)—15% and timothy (*Phleum pratense* L.)—5%, was sown on plots of 27.0 m^2^ (5.4 × 5.0 m) in an amount of 25 kg of seed material per 1 ha.

The experiment was composed of two factors:FGD gypsum (flue gas desulfurization) or synthetic gypsum—two levels (0 and 500 kg CaSO_4_ × 2 H_2_O ha^−1^); referred to as GYP─0 and GYP─500;Phosphorus and potassium treatments—five levels; referred to as AC, P60K0, P60K30, P60K60 and P60K120, respectively.

FGD-gypsum contains calcium sulphate (CaSO_4_ × 2H_2_O) with 21.3% of Ca and 17% of S (authors’ own data). This product was applied before the field experiment was established and each year before the spring regrowth of the sward of alfalfa–grass. Altogether, 2000 kg ha^−1^ of gypsum was used. K as a muriate of potash (60% K_2_O) was applied each year before the beginning of the consecutive growing season. Additionally, in spring, before the beginning of each of the growing seasons, two weeks after the application of calcium sulphate, uniform P fertilization at a dose of 60 kg P_2_O_5_ ha^−1^ (triple superphosphate 46% P_2_O_5_) was applied. Fertilizers for the alfalfa–grass sward were applied as previously discussed.

### 4.4. Plant Sampling and Measurements

The sward of alfalfa–grass was harvested three times during the growing seasons (cuts) at the full budding phase of alfalfa from an area of 12.5 m^2^ (2.5 × 5 m). The sward was weighed and the yield of the dry weight per hectare was calculated. The total yield from the three harvests was summed up. The plant material utilized for nutrient determination were mineralized at 600 °C. The obtained ash was then dissolved in 33% HNO_3_. The concentrations of K, Mg, Ca, Fe, Mn, Zn and Cu were determined using FAAS.

### 4.5. Calculated Parameters

Grass tetany ratio—CR
$$\mathrm{CR}=\frac{K}{\mathrm{Ca}+\mathrm{Mg}}$$Nutrient unit productivity—NUP

$$\mathrm{NUP}=\frac{\mathrm{TY}}{\mathrm{TA}\left(n\right)}\;\mathrm{kg}\;\mathrm{TY},\;g\;\mathrm{grain}\;\times {\mathrm{kg}}^{-1}\;N$$
where:


K, Ca, Mg, Fe, Mn, Zn, Cu—nutrient content in the sward, % DM;TY—total yield of the alfalfa–grass sward, t ha^−1^;TA—total nutrient accumulation, kg or g ha^−1^;n—particular nutrient.


### 4.6. Statistical Analysis

The effect of the experimental factors (year, FGD-gypsum doses, potassium doses) and their mutual interactions on the total yield, the content of nutrients and their accumulation in the alfalfa–grass sward were assessed by an analysis of variance. The means were separated by honest significant difference (HSD) using the Tukey method when the F-test showed significant factor effects at *p* < 0.05. The relationships between the examined characteristics were analyzed using Pearson correlation and linear regression. The stepwise regression analysis was used to determine the optimal set of variables for a given plant trait. The best regression model was selected based on the highest *F*-value for the entire model. STATISTICA 12 software was used for all the statistical analyses (StatSoft Inc., Tulsa, OK, USA, 2013).

## 5. Conclusions

During the period of the study, when the weather conditions were suitable for the growth of alfalfa, it was found that the nutrients had a substantial effect on the yield and nutrient quality of the harvested sward. The highest yield was harvested in the second season of the main use of the sward. The simultaneous application of gypsum and potassium to soil naturally poor in calcium and depleted in potassium caused a significant increase in the total yield of the alfalfa–grass sward. The gypsum application increased the yield by 1.0 t ha^−1^ and potassium led to a net increase in the yield with a maximum of 2.7 t ha^−1^ of 14.9 ha^−1^ at 115 kg K_2_O ha^−1^. The effect of phosphorus was significant, but the total yield of the sward depended on the potassium content and its uptake by plants. In each season, the yield increased progressively with the increase in the amount of K in the sward. Compared to the absolute control, its maximum increase was 67% (162 → 271 kg K ha^−1^), 45% (304 → 441 kg K ha^−1^) and 76% (204 → 360 kg K ha^−1^) in the subsequent seasons, respectively. The total yield was predicted on the basis of four indicators: K content in the first cut of the alfalfa–grass sward, total accumulation of K or Mg in the sward and unit manganese productivity. The K/Ca + Mg ratio as an indicator of feed nutritional quality depended on the year of the sward use, reaching the highest values in the second main season. The application of potassium usually increased this ratio above the typical levels. The last indicator from this group highlighted the limitations of potassium productivity due to a shortage of manganese in the sward. Optimal conditions for growth and yielding of the alfalfa–grass sward created by the use of gypsum and potassium, in turn, caused disturbances in the nutritional status of iron and manganese. As a consequence, a significant shortage of all macronutrients was observed in the sward, especially manganese. The applied gypsum, through increasing the amount of micronutrients in the sward, particularly manganese, increased the productivity of the sward. It can be clearly concluded that even moderate application of macronutrients to an alfalfa-dominated sward must include micronutrients, especially manganese and iron, in the fertilization system of this crop.

## Figures and Tables

**Figure 1 plants-12-02250-f001:**
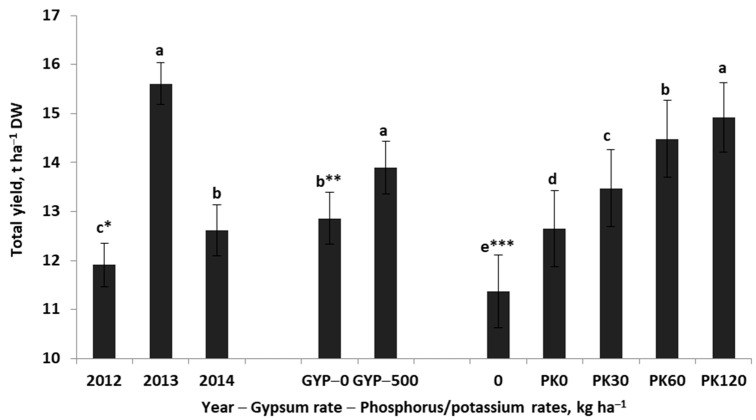
Total sward yield of the alfalfa–grass mixture in response to the years and experimental factors. Similar letters indicate a lack of significant differences using Tukey’s test. The vertical bar in the column refers to the standard error of the mean; *, **, ***—experimental factors: years, gypsum rates, phosphorus/potassium rates, respectively.

**Figure 2 plants-12-02250-f002:**
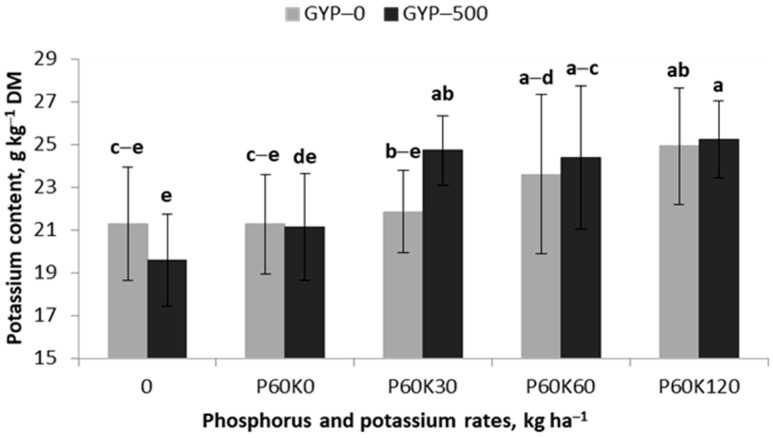
Effect of gypsum and phosphorus/potassium rates on potassium content in the sward of the first cut of alfalfa–grass mixture. Similar letters indicate a lack of significant differences using Tukey’s test. The vertical bar in the column refers to the standard error of the mean.

**Figure 3 plants-12-02250-f003:**
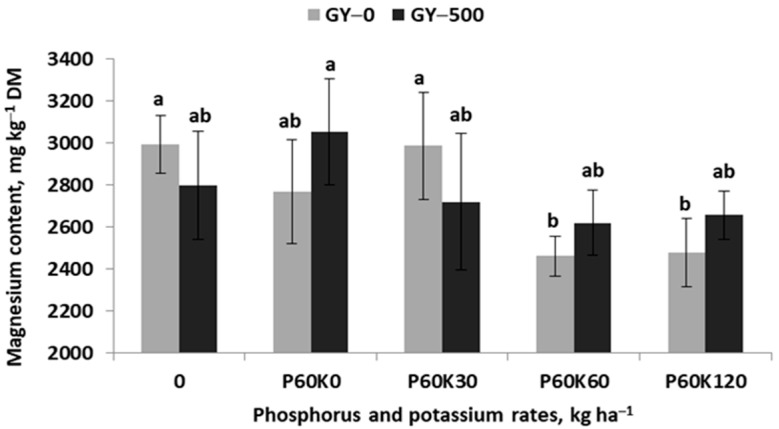
Effect of gypsum and phosphorus/potassium rates on magnesium content in the sward of the third cut of alfalfa–grass mixture. Similar letters indicate a lack of significant differences using Tukey’s test. The vertical bar in the column refers to the standard error of the mean.

**Figure 4 plants-12-02250-f004:**
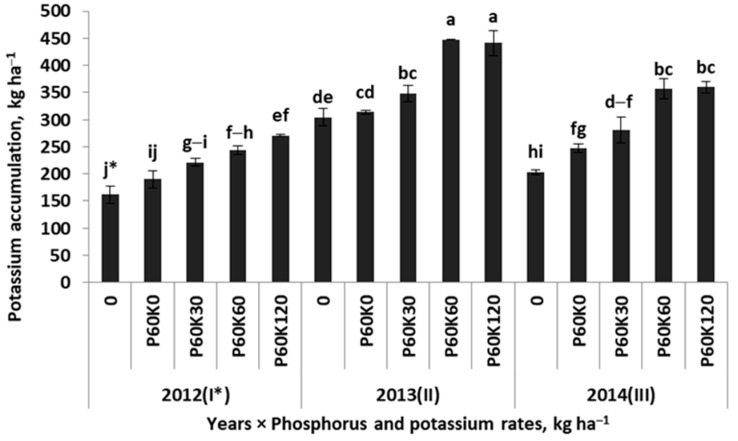
Effect of phosphorus/potassium rates in successive years on the total potassium accumulation in the sward of the alfalfa–grass mixture. Similar letters indicate a lack of significant differences using Tukey’s test. The vertical bar in the column refers to the standard error of the mean; * first main season of the sward use.

**Figure 5 plants-12-02250-f005:**
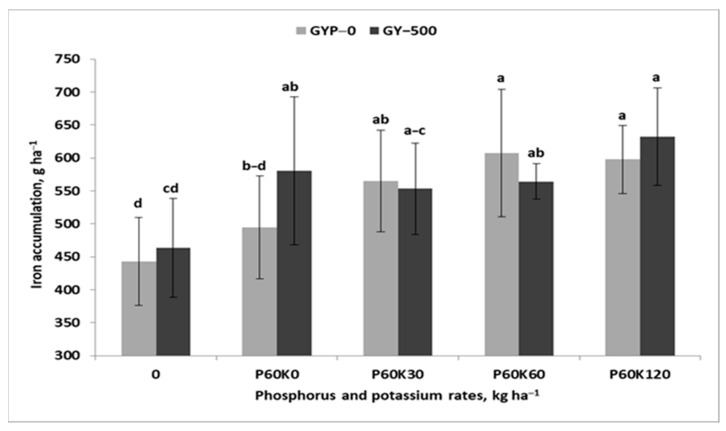
Effect of gypsum and phosphorus/potassium rates on total iron accumulation in the sward of the alfalfa–grass mixture. Similar letters indicate a lack of significant differences using Tukey’s test. The vertical bar in the column refers to the standard error of the mean.

**Figure 6 plants-12-02250-f006:**
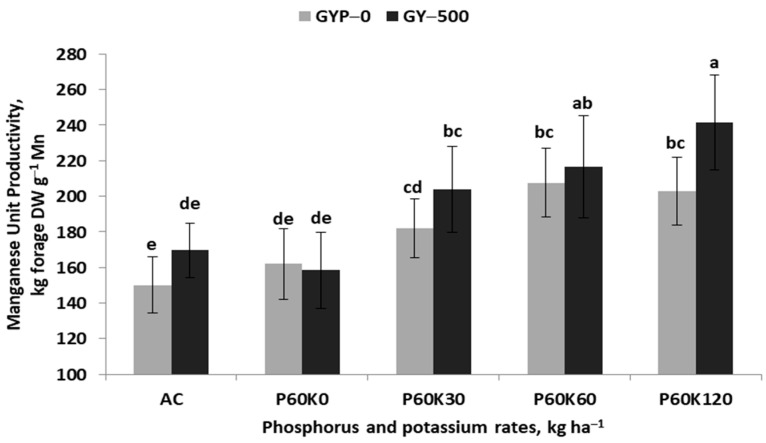
Effect of gypsum and phosphorus/potassium doses on manganese unit productivity in the sward of the alfalfa–grass mixture. Similar letters indicate a lack of significant differences using Tukey’s test. The vertical bar in the column refers to the standard error of the mean.

**Table 1 plants-12-02250-t001:** The content of macronutrients in the sward of the mixture of alfalfa–grass in successive cuts.

Factor	Factor	K1	K2	K3	Mg1	Mg2	Mg3	Ca1	Ca2	Ca3	CR1	CR2	CR3	TY
	Level	g kg^−1^ DM			mg kg^−1^ DM						-			t ha^−1^
Year	2012	20.2 b	18.3 c	15.5 c	3294 a	2351 c	2927 a	7324 b	3204 a	6069 a	2.0 b	4.0 a	1.9 c	11.9 c
(Y)	2013	27.1 a	24.5 a	21.5 b	3396 a	3356 a	2398 b	5792 c	11,198 b	5989 a	3.1 a	1.9 b	2.8 b	15.6 a
	2014	21.1 b	20.7 b	28.6 a	2308 b	2780 b	2932 a	8801 a	13,408 a	3105 b	2.1 b	1.4 c	5.3 a	12.6 b
*p*		***	***	***	***	***	***	***	***	***	***	***	***	***
FGD (GYP)	0	22.6	22.3 a	22.5 a	3013	2841	2737	7334	8725 b	4871	2.3	2.5 a	3.4	12.9 b
kg ha^−1^	500	23.0	20.1 b	21.3 b	2986	2817	2768	7278	9815 a	5237	2.4	2.2 b	3.2	13.9 a
*p*		ns	***	*	ns	ns	ns	ns	*	ns	ns	*	ns	***
PK	0	20.4 b	18.7 c	20.2 b	3124 a	2977 a	2894 a	9051 a	10,656 a	6274 a	1.9 c	2.0 b	2.5 b	11.4 e
Fertilization	PK-0	21.2 b	19.6 bc	19.6 b	3212 a	2949 a	2909 a	8336 ab	8228 b	5260 b	1.9 c	2.2 b	2.6 b	12.7 d
(POT)	PK-30	23.3 a	20.6 b	20.7 b	2827 b	2864 ab	2852	6845 bc	9912 ab	5151 b	2.4 b	2.0 b	2.9 b	13.5 c
	PK-60	24.0 a	24.2 a	24.2 a	2995 ab	2563b	2541 b	6489 c	8130 b	4205 c	2.7 b	3.5 a	4.3 a	14.5 b
	PK-90	25.1 a	22.8 a	24.7 a	2840 b	2791 ab	2567 b	5808 c	9424 ab	4380 bc	3.0 a	2.2 b	4.2 a	14.9 a
*p*		***	***	***	***	**	***	***	**	***	***	***	***	***
Source of variation for the studied interactions														
Y × GYP		**	***	ns	ns	*	ns	ns	ns	***	**	ns	*	ns
Y × POT		***	*	*	ns	***	*	***	ns	***	***	*	***	ns
GYP × POT		*	ns	*	ns	**	*	ns	***	***	*	***	ns	ns
Y × GYP × POT		ns	ns	ns	ns	**	ns	ns	***	***	*	**	ns	ns
Mean		22.8	21.2	21.9	2999.6	2828.9	2752.6	7305.6	9270.0	5054.1	2.4	2.4	3.3	13.4
Standard deviation		4.1	3.8	6.1	543.7	557.2	366.3	2150.6	5196.0	2027.0	0.8	1.5	1.8	2.2
Coefficient of variation, %		18.2	17.9	28.1	18.1	19.7	13.3	29.4	56.1	40.1	32.7	62.2	55.1	16.2

Similar letters in the column indicate a lack of significant differences between experimental treatments using Tukey’s test; ***, **, * indicate significant differences at *p* < 0.001, *p* < 0.01 and *p* < 0.05, respectively; ns—nonsignificant. Legend: Y—years; FGD-GYP—flue gas desulfurization gypsum; K, Mg, Ca—nutrients; 1, 2, 3—successive cuts of the alfalfa–grass sward; CR—K/(Ca + Mg) ratio; TY—total yield of the alfalfa–grass sward.

**Table 2 plants-12-02250-t002:** The accumulation of main macro-cations in the sward of the mixture of alfalfa–grass in successive cuts.

Factor	Factor	K1a	K2a	K3a	Mg1a	Mg2a	Mg3a	Ca1a	Ca2a	Ca3a	TK	TMg	TCa
	Level	kg ha^−1^											
Year	2012	63.8 b	97.3 b	56.3 c	10.3 b	12.3 b	10.6 b	22.7 b	16.9 c	22.1 b	217.4 c	33.1 b	61.7 b
(Y)	2013	105.2 a	101.8 a	164.1 a	13.1 a	13.9 a	18.2 a	22.0 b	45.9 b	44.7 a	371.1 a	45.1 a	112.6 a
	2014	101.1 a	95.4 ab	92.9 b	10.8 b	12.7 b	9.3 c	40.1 a	61.6 a	9.6 c	289.4 b	32.8 b	111.2 a
*p*		***	*	***	***	***	*	***	***	***	***	***	***
FGD (GYP)	0	84.4 b	99.8	104.9	10.8 b	12.5	12.2 b	26.8	37.8 b	24.2 b	289.0	35.6 b	88.8 b
kg ha^−1^	500	95.7 a	96.5	104.0	11.9 a	13.4	13.2 a	29.7	45.1 a	26.7a	296.2	38.4 a	101.5 a
*p*		***	ns	ns	***	*	**	ns	**	*	ns	***	***
PK	0	67.0 c	74.3 c	81.8 c	10.2 c	11.9 b	11.3 b	30.5 a	42.0 ab	27.7 a	223.1 d	33.4 c	100.1
Fertilization	PK-0	75.7 c	86.0 b	88.3 bc	11.2 bc	12.9 b	12.9 ab	30.3 a	35.8 ab	27.0 a	250.0 c	37.0 b	93.1
(POT)	PK-30	92.2 b	94.4 b	96.9 b	10.9 a–c	13.2 b	13.0 a	27.6 a	44.1 ab	25.4 a	283.5 b	37.1 ab	97.1
	PK-60	105.3 a	118.6 a	125.2 a	12.5 a	12.4 ab	12.8 a	27.5 a	38.2 b	23.3 a	349.2 a	37.7 ab	89.0
	PK-90	109.8 a	117.6 a	129.9 a	12.0 ab	14.4 a	13.4 a	25.4 a	47.1 a	23.9 a	357.3 a	39.9 a	96.4
*p*		***	*	***	***	***	**	*	**	*	***	***	ns
Source of variation for the studied interactions													
Y × GYP		ns	***	**	ns	**	ns	ns	ns	**	***	*	ns
Y × POT		***	*	***	ns	***	ns	ns	ns	***	***	**	*
GYP × POT		***	ns	**	ns	**	ns	ns	***	***	ns	ns	ns
Y × GYP × POT		**	ns	ns	ns	***	ns	ns	***	***	ns	ns	***

Similar letters in the column indicate a lack of significant differences between experimental treatments using Tukey’s test; ***, **, * indicate significant differences at *p* < 0.001, *p* < 0.01 and *p* < 0.05, respectively; ns—nonsignificant. Legend: Y—years; FGD-GYP—flue gas desulfurization gypsum; K, Mg, Ca—nutrients; 1, 2, 3—successive cuts of the alfalfa–grass sward; a, T—accumulation. total.

**Table 3 plants-12-02250-t003:** The content of micronutrients in the sward of the mixture of alfalfa–grass in successive cuts.

Factor	Factor	Fe1	Fe2	Fe3	Mn1	Mn2	Mn3	Zn1	Zn2	Zn3	Cu1	Cu2	Cu3
	Level	mg kg^−1^ DM											
Year	2012	40.8 b	41.3 b	33.3 b	23.5 a	30.2 a	23.8 b	34.5 a	22.3 a	19.7	4.8 a	3.8 c	4.3 b
(Y)	2013	63.2 a	52.1 a	30.8 b	23.8 a	29.9 a	17.2 c	25.8 b	18.7 b	20.6	4.3 b	4.4 b	3.3 c
	2014	34.6 c	31.9 c	57.2 a	16.4 b	16.1 b	35.9 a	17.5 c	13.9 c	20.3	5.0 a	5.0 a	4.7 a
*p*		***	***	***	***	***	***	***	***	ns	***	***	***
FGD (GYP)	0	45.8	42.2	42.9	21.1	25.3	25.5b	26.4	18.7	20.3	4.9 a	4.4	4.2
kg ha^−1^	500	46.6	41.3	38.0	21.4	25.5	25.8a	25.5	17.9	20.1	4.5 b	4.4	4.1
*p*		ns	ns	ns	ns	ns	*	ns	ns	ns	***	ns	ns
PK	0	43.7 b	42.3	39.0	21.4 ab	26.1	24.7	25.2 bc	17.9	21.3 a	4.4 b	4.2 b	4.3 a
Fertilization	PK-0	54.5 a	41.4	39.2	24.5 a	27.9	28.2	25.7 bc	18.8	17.9 b	5.0 a	4.3 b	4.0 ab
(POT)	PK-30	45.4 b	42.1	41.1	21.2 ab	23.9	26.1	23.6 c	18.6	19.3 ab	4.7 ab	4.9 a	4.3 ab
	PK-60	42.7 b	41.4	41.6	19.7 b	25.9	23.8	28.0 b	18.1	21.4 a	4.6 ab	4.2 b	3.9 b
	PK-90	44.8 b	41.6	41.3	19.5 b	23.3	25.5	27.4 ab	18.2	21.1 a	4.6 ab	4.3 b	4.2 ab
*p*		**	ns	ns	***	ns	ns	***	ns	**	*	***	*
Source of variation for the studied interactions													
Y × GYP		***	***	ns	*	*	ns	***	*	ns	ns	***	**
Y × POT		***	*	***	ns	ns	ns	***	ns	***	***	***	***
GYP × POT		**	ns	***	ns	ns	ns	***	**	**	*	ns	***
Y × GYP × POT		**	ns	*	ns	ns	ns	**	**	***	*	ns	**
Mean		44.9	41.8	40.4	21.3	25.4	25.7	26.0	18.3	20.2	4.7	4.4	4.1
Standard deviation		12.7	9.9	13.2	4.6	7.5	8.6	7.9	4.1	3.5	0.6	0.9	0.8
Coefficient of variation, %		28.3	23.8	32.6	21.5	29.6	33.4	30.3	22.3	17.4	13.0	19.8	19.8

Similar letters in the column indicate a lack of significant differences between experimental treatments using Tukey’s test; ***, **, * indicate significant differences at *p* < 0.001, *p* < 0.01 and *p* < 0.05, respectively; ns—nonsignificant. Legend: Y—years; FGD-GYP—flue gas desulfurization gypsum; Fe, Mn, Zn, Cu—nutrients; 1, 2, 3—successive cuts of the alfalfa–grass sward.

**Table 4 plants-12-02250-t004:** The accumulation of main micro-cations in the sward of the mixture of alfalfa–grass in successive cuts.

Factor	Factor	Fe1a	Fe2a	Fe3a	Mn1a	Mn2a	Mn3a	Zn1a	Zn2a	Zn3a	Cu1a	Cu2a	Cu3a	TFe	TMn	TZn	TCu
	Level	g ha^−1^															
Year	2012	127.6 b	216.7 a	119.6 c	72.9 b	158.3 a	86.2 c	108.1 a	117.1 a	71.6 b	14.8 b	20.1 b	15.6 b	463.9 b	77.6 a	296.8 b	50.5 b
(Y)	2013	242.3 a	215.0 a	234.7 a	91.1 a	123.4 b	130.5 a	99.9 b	77.2 b	156.1 a	16.5 b	18.1 c	25.1 a	692.0 a	71.0 b	333.2 a	59.7 a
	2014	163.9 c	146.5 b	184.4 b	77.5 b	73.9 c	115.7 b	83.9 b	63.6 c	65.3 b	23.6 a	22.8 a	15.2 b	494.8 b	68.4 b	212.8 c	61.6 a
*p*		***	***	***	***	***	***	***	***	***	***	***	***	***	***	***	***
FGD (GYP)	0	168.5 b	187.4	185.7 a	75.5 b	112.8 b	107.0	94.3 b	84.0	95.7	18.1	19.6 b	18.5	541.7	71.9	273.9 b	56.3
kg ha^−1^	500	187.4 a	198.0	173.4 b	85.5 a	124.3 a	114.6	100.3 a	88.0	99.7	18.4	21.1 a	18.8	558.8	72.7	288.0 a	58.3
*p*		*	ns	*	***	***	ns	*	ns	ns	ns	*	ns	ns	ns	***	ns
PK	0	141.9 b	167.9 c	143.5 b	68.7 b	104.0 b	92.6 a	80.3 b	70.8 c	90.4 b	14.5 c	16.9 c	16.7 b	453.3 c	72.1 b	241.5 c	48.1 c
Fertilization	PK-0	193.8 a	181.5 bc	162.4 b	85.8 a	124.9 ab	117.4 a	86.3 b	84.2 b	82.5 b	17.6 b	19.2 bc	17.6 b	537.7 b	80.6 a	252.9 bc	54.4 b
(POT)	PK-30	178.6 a	192.8 a–c	187.9 a	81.5 ab	111.5 ab	111.2 ab	90.4 b	87.8 ab	90.3 b	19.1 ab	23.0 a	18.6 b	559.3 ab	71.1 b	268.4 b	60.7 a
	PK-60	181.9 a	204.3 ab	199.8 a	82.5 a	129.4 a	111.1 ab	114.2 a	90.7 ab	116.5 a	19.8 ab	20.3 ab	19.0 ab	586.0 ab	69.4 b	321.4 a	59.1 ab
	PK-90	193.6 a	217.1 a	204.3 a	84.0 a	123.0 ab	121.6 a	115.2 a	96.5 a	108.8 a	20.5 a	22.4 a	21.3 a	614.9 a	68.4 b	320.4 a	64.1 a
*p*		***	***	***	***	*	**	***	***	***	***	***	***	***	***	***	***
Source of variation for the studied interactions																	
Y × GYP		**	*	*	ns	***	ns	**	***	ns	ns	***	***	ns	*	***	***
Y × POT		***	*	***	ns	*	*	**	***	***	***	***	***	ns	**	***	***
GYP × POT		ns	**	***	ns	ns	ns	***	**	***	ns	***	***	*	*	***	*
Y × GYP × POT		**	**	*	ns	ns	ns	ns	**	***	ns	**	ns	ns	ns	***	ns

Similar letters in the column indicate a lack of significant differences between experimental treatments using Tukey’s test; ***, **, * indicate significant differences at *p* < 0.001, *p* < 0.01 and *p* < 0.05, respectively; ns—nonsignificant. Legend: Y—years; FGD-GYP—flue gas desulfurization gypsum; Fe, Mb, Zn, Cu—nutrients; 1, 2, 3—successive cuts of the alfalfa–grass sward; a, T—accumulation total.

**Table 5 plants-12-02250-t005:** Indices of nutrient unit productivity–NUP in the sward of the mixture of alfalfa–grass.

Factor	Factor Level	K	Mg	Ca	Fe	Mn	Zn	Cu
		kg Fodder DM kg^−1^			kg Fodder DM g^−1^			
Year (Y)	2012	55.7 a	363.7 b	200.3 a	25.8 b	156.6 c	40.6 c	240.7 b
	2013	43.1 b	347.4 b	146.6 c	23.0 b	225.0 a	47.8 b	264.3 a
	2014	44.9 b	386.8 a	117.5 b	26.1 a	186.5 b	60.5 a	209.1 c
*p*		***	***	***	***	***	***	***
FGD (GYP) kg ha^−1^	0	47.1 b	366.0	160.1 a	24.4 b	180.8 b	48.7 b	233.8 b
	500	48.7 a	365.8	149.4 b	25.6 a	197.9 a	50.6 a	242.2 a
*p*		*	ns	*	*	***	**	***
PK Fertilization (POT) kg ha^−1^	AC	53.2 a	345.1 b	134.8 c	25.8	159.8 c	49.9 ab	240.1 ab
	PK-0	51.8 a	346.8 b	149.5 bc	24.4	160.1 c	52.0 a	236.2 ab
	PK-30	48.6 b	369.3 ab	148.1 bc	24.9	192.8 b	52.0 a	227.5 b
	PK-60	43.0 c	392.0 a	176.9 a	25.1	212.1 a	46.8 c	248.7 a
	PK-120	42.8 c	376.5 a	164.6 ab	24.6	222.1 a	47.4 bc	237.7 ab
*p*		***	***	***	ns	***	***	*
Source of variation for the studied interactions								
Y × GYP		**	*	*	ns	*	***	ns
Y × POT		*	**	ns	*	*	***	ns
GYP × POT		ns	ns	ns	*	*	***	ns
Y × GYP × POT		ns	ns	*	ns	ns	***	ns

Similar letters in the column indicate a lack of significant differences between experimental treatments using Tukey’s test; ***, **, * indicate significant differences at *p* < 0.001, *p* < 0.01 and *p* < 0.05, respectively; ns—nonsignificant. Legend: Y—years; FGD-GYP—flue gas desulfurization gypsum; K, Mg, Ca, Fe, Mn, Zn, Cu—nutrients.

**Table 7 plants-12-02250-t007:** Weather conditions during the vegetation period in experimental station Brody in the years 2012–2014.

Month	Average Air Temperature (°C)				Total Rainfall (mm)			
	2012	2013	2014	1961–2011	2012	2013	2014	1961–2011
IV	8.8	8.0	10.5	8.0	22.9	15.4	46.3	37.6
V	14.9	14.4	13.1	13.2	77.2	69.8	73.5	56.9
VI	16.0	17.3	16.1	16.6	163.0	125.3	42.0	61.6
VII	19.2	20.1	21.5	18.2	197.6	67.3	83.1	79.4
VIII	18.7	19.1	17.3	17.5	60.1	51.5	137.2	66.9
IX	14.3	12.9	15.4	13.3	30.0	33.7	64.8	49.7
X	8.2	10.3	10.9	8.5	10.9	10.9	39.8	40.8
IV–X average temperature	14.3	14.6	15.0	13.6	-	-	-	-
IV–X rainfall	-	-	-	-	561.7	373.9	486.7	392.9
Annual average temperature	8.9	8.8	10.1	-	-	-	-	-
Annual rainfall	-	-	-	-	811.5	516.5	632.5	-

## Data Availability

Not applicable.

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
