# Peer review of "Mutual Effect of Gypsum and Potassium on Nutrient Productivity in the Alfalfa–Grass Sward—A Case Study"

_plants, 2023, doi:10.3390/plants12122250_

Round 1

Reviewer 1 Report

I suggest major revision

Author Response

Review report 1 _ response

ID:plants-2399596

The authors of this manuscript thank the reviewer for comments that have increased its scientific value.

Specific comments

Responses to the reviewer comments are marked in red

Introduction
Line 98: remove “should”: “..calcium fertilizers

This has been corrected.

Line 109: “Yield was 3.6 t ha-1 (31%) higher than that of the first main year of use (2012)....”

Average sward yield this season was by 3.6 t ha─1 (31%) higher compared to the first main year of use (2012)….
Line 110: “Gypsum application resulted in a Total Yield (TY) increase of...”

The application of gypsum, averaged over the years and PK fertilization resulted in a TY increase of 1.0 t ha─1 (7.8%).

Line 116: “...+ 12.6 for n = 30, R2 = 99,....”

This has been corrected

Line 134-5: “..on the plot with 60 kg K ha-1 (Table 1). Despite this, the growth trend can be
best described using a linear regression model (Figure 2....

The content of K in the sward on GYP─0 (gypsum control) showed a stable level on the plots with no or a low dose of K, while its significant increase was recorded for the first time on the plot with 60 kg K ha─1 (Table 1). Nevertheless, the trend of K content on plots fertilized with K was consistent with the linear regression model (Figure 2, R2 = 0.88).

Section 2.3
Line 53: What do you mean by Fe being twice as low in the third cut? In 2012 the Fe decline from 40.8 to 33.3 both have “b” so they are not significantly different? I suggest: “The content of Fe in each cut varied greatly between seasons: while there was no significant change between cuts during 2012, there was a decrease during 2013, and an increase during 2014.”

This passage has been corrected as suggested by the reviewer.

Page 2 of 34: “The average content of Mn was 24.1+2.5, and during both 2012 and 2013 Mn was highest in the second cut, though in 2013, as with Fe, it was highest in the third cut (Table 3). The observed trends....”

This passage has been corrected as suggested by the reviewer.

2nd paragraph: The statement that the same trend for Fe was observed with Cu is NOT true. While the average content of Cu was 4.4+0.3 mg kg-1 DM, the Cu content was highly variable between cuts. Cu content was lowest in the second cut during 2012, in the third cut in 2013, while in 2014, it was higher. The use of gypsum resulted in a significantly lower Cu content in the sward of the first cut (Table 3).

I agree with the reviewer. This passage has been corrected as suggested by the reviewer.

Section 2.4
Near bottom of page 2 of 34: “..had the greatest impact on Total Yield (TY):..”

Top of page 3 of 34: if effect of gypsum on K plot with 120 was not significant, then it is NOT clear so omit this statement.

This passage has been corrected as suggested by the reviewer.

Discussion
Far too many numbers repeated in the discussion. These are a repeat of the results.

Some of the results were included intentionally to indicate that good production effects were obtained in conditions of significant and lower expenditures, especially for K doses, than it is in the literature circulation.

So:
Line 3: “The total yield ....was high, reflecting the optimal precipitation during the study period. The total yield as harvested in the second season....”
There is no need to mention r values or Tables where data is found: these are results. So:

These items have been removed from the discussion.

Line 38-39: “The limiting effect of calcium on the productivity of K was small, with
manganese being most important.”
This passage has been corrected as suggested by the reviewer.

Line 22: “..driven by progressive doses of applied potassium. The increase in Total Yield (TY) by .....”
This passage has been corrected as suggested by the reviewer.

Line 55: “The importance of K fertilizer on yield was enhanced...”
This passage has been corrected as suggested by the reviewer.

Line 74: Reduce to: “An important relationship is the tetany ratio, between K on the one hand and Ca and Mg on the other and this ratio is optimal in the range of 2.0-2.2 [25;28]. This nutritional index..”
This passage has been corrected as suggested by the reviewer.

Lines 80-86 as a sentence: “The calculated ranges for the amount of Ca required in the sward of the alfalfa-grass mixture for the first to third cuts were 0.73-0.80, 0.68-0.78 and 0.60-0,66 mg kg-1 DM respectively, indicating the highest requirements for Ca during the spring growth period. The most important effect of gypsum...”
This passage has been corrected as suggested by the reviewer.

Materials and Methods
Mention the size of each plot.
The sward of alfalfa-grass was harvested three times during the growing seasons (cuts) at the full budding phase of alfalfa from an area of 12.5 m2 (2.5 × 5 m).

Line 144: “increasing gradually from 6.5 to 6.7 to 6.9, in soil layers....”

It has been corrected.

“Soil pH was in the neutral range (1 M KCl) but increased gradually with soil depth.”

Line 146-147: Use very high to very low only, omit excellent to bad.
It has been corrected.

Line 171-172: give species names of alfalfa, meadow fescue and timothy.
It has been corrected.

Conclusions

After a very long discussion, which was far too detailed, the authors finally come to some conclusions as to what to recommend to farmers. But let’s make it stronger by separating the effects of each important nutrient:

“During the period of the study, when weather conditions were suitable for the growth of alfalfa, it was found that nutrients had a substantial effect on yield and nutrient quality of the harvested sward. Simultaneous application of gypsum and potassium....total yield of the alfalfa-grass sward. Gypsum application increased the yield by 1.0 t ha-1 and potassium led to an increase in yield with a maximum of .... At 115 kg....” The effect of phosphorus was significant in that the total yield depended on the potassium content and uptake by the plants. Total yield was predicted on the basis of four indicators: K content ....”

This passage has been corrected as suggested by the reviewer.
Line 237: “..macronutrients to an alfalfa dominated sword must include micronutrients,...”

This passage has been corrected as suggested by the reviewer.

On behalf of the Authors

Professor Dr. Witold Grzebisz

Reviewer 2 Report

The most recent article, "Mutual Effect of Gypsum and Potassium on Nutrient 2 Productivity in the Alfalfa-Grass Sward – A case Study " combines knowledge with application. Readers of the journal and other researchers working on the topic internationally may find this publication useful. The manuscript was meticulously written. This is a significant effort that merits praise. I think that after some minor revisions, this work should be published in light of the following points.

1.      In the manuscript title provide the botanical name of the plant.

2.      Stick to the journal's reference format for reference and provide all possible DOI numbers.

3.      Apart from all the above-mentioned comments, authors are also asked to recheck the whole manuscript for any grammatical or typical errors in the revised version. 

4.      Add more recent references.

5.      Provide conclusions by including more significant results.

The most recent article, "Mutual Effect of Gypsum and Potassium on Nutrient 2 Productivity in the Alfalfa-Grass Sward – A case Study " combines knowledge with application. Readers of the journal and other researchers working on the topic internationally may find this publication useful. The manuscript was meticulously written. This is a significant effort that merits praise. I think that after some minor revisions, this work should be published in light of the following points.

1.      In the manuscript title provide the botanical name of the plant.

2.      Stick to the journal's reference format for reference and provide all possible DOI numbers.

3.      Apart from all the above-mentioned comments, authors are also asked to recheck the whole manuscript for any grammatical or typical errors in the revised version. 

4.      Add more recent references.

5.      Provide conclusions by including more significant results.

Author Response

Review report 2_response

Comments and Suggestions for Authors

The most recent article, "Mutual Effect of Gypsum and Potassium on Nutrient 2 Productivity in the Alfalfa-Grass Sward – A case Study " combines knowledge with application. Readers of the journal and other researchers working on the topic internationally may find this publication useful. The manuscript was meticulously written. This is a significant effort that merits praise. I think that after some minor revisions, this work should be published in light of the following points.

Responses to the reviewer comments are marked in red

  1. In the manuscript title provide the botanical name of the plant.

A mixture consisting of alfalfa and two species of grasses, meadow fescue and timothy, was used in the experiment. The share of grass species in the sward did not exceed 15% in the sowing season, and in the third year of use it was less than 10%. The introduction of grass names to the title will cause a kind of information chaos. The botanical names of the tested species are given in the Materials and Methods section.

  1. Stick to the journal's reference format for reference and provide all possible DOI numbers.

The correctness of the cited literature was checked and DOI numbers were added.

  1. Apart from all the above-mentioned comments, authors are also asked to recheck the whole manuscript for any grammatical or typical errors in the revised version. 

Stylistic, grammatical error and so-called typos have been corrected. English editing of the manuscript has been done by an English lecturer at the Adam Mickiewicz University in Poznan, Mr. Robert Kippen.

  1. Add more recent references.

In terms of conducted research, it is easier to find items from the 20th century that at present. Four new items were added. In the corrected reference list, 24 items (50%) are younger than 2010 and 11 items represents the last four years.

  1. Provide conclusions by including more significant results.

The conclusions have been corrected. Significant results are included.

On behalf of the Authors

Professor Dr. Witold Grzebisz

Reviewer 3 Report

This paper presents a lot of detail and appears well conducted especially the analyses. I think the methods should include a description of how the fertilizers were applied (e.g. broadcast or drilled) as this may affect how rapidly and well the nutrients are taken up. 

There are numerous misspellings/typographical errors in the body of the paper. 

I did not have a problem with the English used here.

Author Response

Review report  3_response

Comments and Suggestions for Authors

Responses to the reviewer comments are marked in red

This paper presents a lot of detail and appears well conducted especially the analyses. I think the methods should include a description of how the fertilizers were applied (e.g. broadcast or drilled) as this may affect how rapidly and well the nutrients are taken up. 

A corresponding correction has been made and reads as follows:

“Fertilizer for the alfalfa-grass sward were applied as broadcast.”

There are numerous misspellings/typographical errors in the body of the paper. 

Stylistic, grammatical error and so-called typos have been corrected. English editing of the manuscript has been done by an English lecturer at the Adam Mickiewicz University in Poznan, Mr. Robert Kippen.

On behalf of the Authors

Professor Dr. Witold Grzebisz

Round 2

Reviewer 1 Report

The authors have accepted most of my suggestions and I accept the responses for the rest.  I found several small errors: Section 2.3: Second paragraph page 2 of 35: The average content of Cu (4.4±0.3 mg kgDM). The cu content was  “Cu content” NOT “cu content”                                                                    And in Discussion Line 74: “tetany ratio” NOT “tetany ration”

Only a few minor corrections needed

Author Response

Review Report (Round 2)_response

The authors have accepted most of my suggestions and I accept the responses for the rest.  I found several small errors: Section 2.3: Second paragraph page 2 of 35: The average content of Cu (4.4±0.3 mg kg─1 DM). The cu content was  “Cu content” NOT “cu content”                                                                    And in Discussion Line 74: “tetany ratio” NOT “tetany ration”

Response

The reviewer’s comments have been taken into account. The text has been corrected.

The correctness of the English language was checked by a teacher  (an Englishman) from Adam Mickiewicz University in Poznań, Poland.

On behalf of the Authors

Professor Dr. Witold Grzebisz
